# The Moderating Effect of Contact with Children on the Relationship between Adverse Childhood Experiences and Depression in Adulthood among a Chinese Adult Population

**DOI:** 10.3390/ijerph19158901

**Published:** 2022-07-22

**Authors:** Yufeng Zhao, Dianxi Wang, Feilun Du

**Affiliations:** 1School of Sociology and Ethnology, University of Chinese Academy of Social Sciences, Beijing 102488, China; 20210001@ucass.edu.cn; 2Institute of Social Development, Chinese Academy of Macroeconomic Research, Beijing 100038, China; dufeilun@sina.com; 3School of Marxism, Beijing Sport University, Beijing 100084, China

**Keywords:** adverse childhood experiences, depression, contact with children, adulthood

## Abstract

The effect of adverse childhood experiences (ACEs) on depression in adulthood has been identified in many studies; however, the underlying mechanisms remain unclear. To understand the moderating effect of ACEs on depression, a moderation analysis using the interaction effect model was performed based on data obtained from the China Health and Retirement Longitudinal Study. This study found that people with ACEs had significantly lower depression scores than those without ACEs, particularly in categories such as physical abuse, emotional neglect, sibling death, parental illness/disability, parental depression, hunger, violence, and bullying. In addition, the results indicated that contact with children moderated the relationship between ACEs and depression in adulthood. Increased levels of contact with children reduced the adverse effects of parental drug abuse and the experience of starvation, but not physical abuse. This study highlights the role of family support in eliminating health disparities, which can reduce the effects of ACEs on depression in adulthood.

## 1. Introduction

Substantial empirical evidence from multiple countries has demonstrated that adverse childhood experiences (ACEs) are highly correlated with depression in adulthood [1,2,3]. For example, scholars have long believed that ACEs may enhance vulnerability to psychological development throughout the course of a person’s life, specifically depression [4,5,6,7,8]. In addition, some meta-analyses confirm that a history of ACEs significantly increases the risk of depression and anxiety in adulthood [9,10,11,12,13]. These results underscore the importance of evaluating the negative effects of ACEs.

The existing literature has also examined the effects of different types of ACEs on depression. Of the various ACEs in the literature, childhood physical and sexual abuse and a history of family violence showed the strongest relationships with anxiety disorders later in life [10,12,14]. Researchers found that childhood verbal and emotional abuse increased the risk of depression [15,16,17]. Sachs-Ericsson et al. [18] also found people who were verbally abused had 1.6 times as many symptoms of depression and anxiety as those who had not been verbally abused and were twice as likely to have experienced a mood or anxiety disorder in their lifetime. Adolescents who experienced peer isolation and emotional neglect were found to be most at risk of symptoms of depression and anxiety [19]. Symptom severity and major depressive episodes were greater for those who experienced community violence and family distress [20]. In addition, Cavanaugh and Nelson [21] found that Black American adults who reported any of the five types of child abuse/neglect or any of the five types of family dysfunction had a greater risk of major depression in the past year.

Some scholars have discussed the influencing mechanisms and factors alleviating the effect of ACEs, as many individuals experience ACEs without developing depression and other psychological problems. Many studies have identified multiple factors that reduce the effect of ACEs on depression. For example, some scholars have found that positive childhood experiences at home and school protect at-risk adolescents against mental health problems [22]. Mindfulness is a cognitive resource that protects people against symptoms of psychological distress [23]. Self-esteem mediates the negative correlation between child maltreatment and depressive symptoms and protects against the adverse effects of early adversity exposure on mental health [24]. Researchers have also found that resilience moderates the association between ACEs and depression, and that the association between ACEs and depression is stronger in low-resilience individuals than in high-resilience individuals [25]. Some scholars have found that religious service attendance and ethnic identity are protective factors for the effects of ACEs on depression [21].

Moreover, some studies revealed that the effect of ACEs on depression may be reduced by managing current stressors and building psychological resilience [26]. Interventions that focus on improving emotion-regulation skills might provide an efficient “transdiagnostic” treatment strategy for psychological and physical health problems [27]. A longitudinal study on acculturation stress and the relationship between ACEs and depression in a sample of young adult Hispanic immigrants indicated that ACEs predicted depressive symptoms and revealed significant mediating and moderating effects of cumulative and distinct facets of acculturation stress [28].

In China, children’s experiences are gradually being diversified due to social transformation. In 2021, the number of rural migrant workers in cities reached 290 million [29]. Children left behind in rural areas are often looked after by their grandparents, while some children move to cities with their parents and face a life of displacement [30,31]. Furthermore, the family structure in China is changing, and the divorce rate is increasing annually; the crude divorce rate rose from 1.1% in 1987 to 3.1% in 2020 [32,33]. Moreover, as the divorce rate rises, the phenomenon of remarriage also increases; data show that the remarriage rate after divorce reached 60%, and the divorce rate after remarriage was as high as 80% in 2020 [33]. Evidently, the proportion of children who suffered from family changes increased, and it was found that children in single-parent households or restructured families were more likely to have negative family experiences [34]. Thus, as Chinese families diversify, the effect of ACEs on adulthood becomes particularly important.

In summation, the existing literature analyzes various aspects of the relationship between ACEs and mental health outcomes in adulthood, revealing that adverse experiences in early life are likely to cause depression in adulthood. However, the mechanisms and mitigating factors of the effect of ACEs on depression in adults remain unclear. Some studies examined the effects of intergenerational transmission of ACEs and found that higher parental childhood adversity scores were associated with higher children’s depression [35], while parents’ positive childhood experiences counteract intergenerational ACEs [36]. However, the bridging effect of intergenerational communication on mental health disadvantages caused by ACEs has not yet been established. Identifying this effect could facilitate a better understanding of the role of ACEs on adult mental health outcomes. Therefore, this study focuses on the effects of ACEs in adult depression, particularly emphasizing the inhibitory effect of contact with children on adult depression. The following research hypotheses were established:

Research hypothesis 1: ACEs have a significant effect on depression, and adults who experienced ACEs are at a higher risk of depression than those who did not experience ACEs.

Research hypothesis 2: Contact with children mitigates the effects of ACEs on depression in adulthood, and communication with their children reduces the risk of developing depression in some individuals who experienced ACEs.

## 2. Materials and Methods

### 2.1. Data

This study used data from the China Health and Retirement Longitudinal Study (more detailed information about CHARLS can be found at http://charls.pku.edu.cn/zh-CN (accessed on 3 May 2022)) conducted by the National School of Development at Peking University. The survey adopted the method of multi-stage sampling. Respondents were selected from 28 provinces (autonomous regions and municipalities) in China in 2011, and follow-up surveys were conducted in 2013, 2014, 2015, and 2018. Information on the respondents’ ACEs was obtained from the 2014 survey. Respondents in this survey retrospectively reported all ACEs, such as physical abuse, emotional neglect, experiencing parental death, divorce, and substance abuse. In addition, the 2015 follow-up survey provided data on mental health in adulthood, and we pooled the two parts of the variables together. Therefore, this survey provided a reliable database to analyze the effect of ACEs on individuals’ depression. After excluding cases with missing data, the final valid sample for this study was 11,564. The average age of respondents was 57.49, with the youngest being 30 and the oldest being 84. The distribution of the selected samples is shown in Table 1.

### 2.2. Variables

#### 2.2.1. Dependent Variable

The dependent variable of this study was the respondents’ depression scores in adulthood based on the depression scores at the time of the survey. This study used the Depression Scale (CES-D) developed by Sirodff [37], which is a self-report scale designed to measure depressive symptomatology in the general population. Depression in adulthood was measured using a 10-item scale, including “I was bothered by things that don’t usually bother me”, “I had trouble keeping my mind on what I was doing”, “I felt depressed”, “I felt everything I did was an effort”, “I felt hopeful about the future”, “I felt fearful”, “My sleep was restless”, “I was happy”, “I felt lonely”, and “I could not get ‘going’.” The answers were rated on a four-point Likert scale (1 = “rarely or none of the time”; 4 = “most or all of the time”). Cronbach’s alpha coefficient in this study was 0.795. The depression score was the total score of the above 10 items, with a minimum value of 2 and a maximum value of 40. Respondents with higher scores indicated higher levels of depression.

#### 2.2.2. Independent and Control Variables

The independent variable was ACEs. Previous studies often used retrospective indicators that allow respondents to recall ACEs that occurred during childhood [38,39]. This study measured the following types of ACEs: physical abuse, emotional neglect, parental divorce, parental death, sibling death, parental drug abuse, parental alcoholism, parental crime, parental illness/disability, parental depression, the experience of starvation, the experience of violence, and being bullied by schoolmates. All questions were dichotomous, with “Yes/No” response options to indicate whether the adverse event happened during childhood.

The control variables included sex (male = 1; female = 2), Hukou type (rural = 1; urban = 2), education level (elementary and below = 1; middle school = 2; high school = 3; college or above = 4), ethnic group (minority = 1; Han = 2), religious belief (no religion = 1; religious affiliation = 2), political status (non-party member = 1; party member = 2), and age (continuous variable).

#### 2.2.3. Moderating Variable

Contact with children was included in the model as a moderating variable measured by the question “How often do you have contact with your own children either by phone, text message, mail, or email?” Options for response were “Almost every day”, “2–3 times a week”, “Once a week”, “Every two weeks”, “Once a month”, “Once every three months”, “Once every six months”, “Once a year”, and “Almost never”. Higher values indicated more frequent contact with children. The maximum value was 9, and the minimum value was 1.

### 2.3. Statistical Analysis

This study used a *t*-test and an F-test to examine differences in depression scores among populations with different characteristics and people with and without ACEs. Using depression scores as the dependent variable, ordinary least squares (OLS) regression was applied to analyze the effect of childhood ACEs on depression scores in adulthood. Moreover, to examine the moderating effect of contact with children on ACEs and depression outcomes during adulthood, this study established the interaction of ACEs and contact with children. Model 1 included only control variables; Model 2 was the main effect model; Model 3 was the interaction effect model. All statistical analyses were conducted using STATA 15.0 (StataCorp LLC, College Station, TX, USA).

## 3. Results

### 3.1. Differences in Depression among People with Different Characteristics

Table 2 shows the average depression scores for different characteristic groups. Female depression scores were significantly higher than male scores. The depression scores of rural residents were significantly higher than those of urban residents. Respondents with a level of education of elementary school and below had the highest depression scores, followed by middle school, whereas respondents with a college or above education background had the lowest depression scores. The depression scores of minority ethnic groups were slightly higher than those of Han ethnic groups. However, there was no significant difference in depression scores between residents with and without religious beliefs and between party members and non-party members.

Table 3 presents the average depression scores for male and female respondents with different ACEs. We found that respondents who experienced physical abuse, emotional neglect, parental death, sibling death, parental illness/disability, parental depression, starvation, violence, and bullying had higher depression scores than those who did not. Male respondents with parental alcoholism and parental criminality scored significantly higher than those without these experiences. Female respondents who experienced parental divorce scored higher on depression without this experience. In addition, parental drug abuse was not significant among male or female respondents.

Table 4 presents the average depression scores for urban and rural residents. Among both urban and rural residents, those who experienced physical abuse, emotional neglect, parental depression, starvation, violence, and bullying had higher depression scores than those who did not. However, parental divorce, parental drug use, and parental alcoholism were not significant among either urban or rural residents. Among urban residents, only parental depression, starvation, violence, and bullying were significant, whereas among rural residents, physical abuse, emotional neglect, parental death, sibling death, parental crime, parental illness/disability, parental depression, starvation, violence, and being bullied were significant.

### 3.2. The Effects of ACDs on Depression

Table 5 presents the results of the OLS regression analysis. Model 1 only included control variables, Model 2 included independent variables of ACEs on the basis of control variables, and Model 3 included the interaction term of ACEs and contact with children. From Model 1 to Model 3, control variables—namely sex, Hukou type, age, religious belief, political status, and education level—had significant effects on respondents’ depression scores.

Among the independent variables in Model 2, the effects of physical abuse, emotional neglect, sibling death, parental illness/disability, parental depression, the experience of starvation, the experience of violence, and being bullied by schoolmates were significant. Specifically, respondents who experienced physical abuse had a depression score that was 0.565 higher than those who did not; those who experienced emotional neglect had depression scores 0.43 higher than those who did not; those who experienced the death of a sibling had depression scores 0.406 higher than those who did not; those whose parents were sick or disabled had depression scores 1.237 higher than those whose parents were not; those whose parents were depressed had depression scores 2.168 higher than those whose parents were not; those who experienced starvation in childhood had depression scores 0.958 higher than those who did not; those who experienced violence had depression scores 0.991 higher than those who did not; those who experienced bullying by classmates had depression scores 1.230 higher than those who did not.

In Model 3, after incorporating the interaction items of ACEs and contact with children, parental illness/disability, parental depression, the experience of starvation, the experience of violence, and being bullied by schoolmates remained significant; however, physical abuse, emotional neglect, and sibling death were no longer significant. The interaction terms of physical abuse, parental drug abuse, the experience of starvation, and contact with children were significant. For every one-unit increase in the frequency of contact with children, those who experienced physical abuse had depression scores 1.862 lower than those who did not. This suggests that contact with children had a significant moderating effect on the depression scores between those who experienced physical abuse and those who did not. We calculated the marginal effect to further show the moderating effect of contact with children, as shown in Figure 1. We found that with increased frequency of contact with their own children, those who were physically abused had significantly higher depression scores than those who were not. In other words, the increased frequency of contact with their children did not eliminate the difference in the depression scores between these individuals.

For parental drug abuse, for every one-unit increase in the frequency of contact with children, respondents whose parents used drugs scored 0.941 higher on depression than those whose parents did not. Figure 2 shows the moderating effect of contact with children on depression scores between people whose parents were drug users and those whose parents were not. As the frequency of contact with their children increased, the depression scores of those whose parents used drugs reduced more than those whose parents did not. In other words, increased bonding with their children contributed more to the improvement in depressive outcomes for those individuals whose parents were drug users than for those whose parents were non-drug users.

For every one-unit increase in the frequency of contact with children, respondents who experienced starvation in childhood scored 0.116 higher on depression than those who did not. Figure 3 shows the marginal effect of the interaction term between experiencing starvation and contact with their children. Increased bonding with their children was more beneficial for reducing depression among respondents who experienced childhood starvation than those who did not. In addition, the interaction terms for parental divorce, parental death, and contact with one’s children were marginally significant. Increased connection with their children improved depression among people who experienced parental divorce and death (see Figure A1 and Figure A2).

## 4. Discussion

This study examined the effect of ACEs on depression and the moderating role of contact with children based on CHARLS data. First, this study further confirmed the fact that most ACEs contribute to depression in adulthood. The present study examined 13 types of ACEs. Eight of these were significant, accounting for 62%. More specifically, adults who experienced physical abuse, emotional neglect, sibling death, parental illness/disability, parental depression, starvation, violence, and being bullied by schoolmates in childhood were more likely to experience depression in adulthood than those who did not. This finding was consistent with extant studies [40,41,42].

Second, the current study demonstrated that not all ACEs cause depression in adulthood and further illustrated that different ACEs have different outcomes. Although the ever-increasing literature indicates that ACEs may trigger mental health problems, such as depression [2,4], not all individuals who experience ACEs develop depression. Five ACEs (parental divorce, parental death, parental drug abuse, parental alcoholism, and parental crime) did not cause depression in adulthood. However, this does not mean that these ACEs did not cause other harm. For instance, parental divorce has been shown to lead to lower self-rated health [43]. The reasons and mechanisms behind only certain ACEs leading to depression and not others require further research.

Third, this study determined the role of contact with children in mitigating the effects of ACEs on depression risks in adulthood. The results revealed that an increase in contact with their children resulted in lower depression scores in adulthood among individuals who experienced parental drug abuse and starvation in childhood, indicating that contact with children mitigated these adverse effects. Childhood adversity should be healed in adulthood, and the family is an important arena for such healing. Contact with children helps ACE victims experience the family warmth that they did not experience as children, thereby reducing the risk of depression. Starvation is a physical experience, and it develops into a psychological experience as an individual grows, evolving into a fear of starvation and a sense of insecurity. Contact with children enables individuals suffering from hunger to gain a sense of security through interaction with family members. Some scholars have noted that resilience [25] and stress management [26] as moderating variables are particularly important when it comes to reducing the effect of ACEs on depression. However, resilience and stress management are both intrinsic qualities of the individual and do not reveal how family intergenerational relationships affect people’s mental health. This study revealed that contact with children moderates the relationship between ACEs and depression, further underscoring the role of family intergenerational relationships in reducing the vulnerability effect of ACEs.

The effects of ACEs on adult depression further indicated the importance of family support. With the moderation of contact with children, parental divorce and death had significant marginal effects. Parental divorce and death are family issues. As mentioned above, Chinese families are becoming increasingly diverse, with rising divorce rates and an increase in single-parent families. The results of this study demonstrated that better communication with their children could reduce the impact of parental divorce and death on depression in an individual’s adult life. In other words, pain caused by the family should be treated by the family. Furthermore, this sheds light on the importance of family policy in reducing the negative impact of childhood adversity [44,45,46]. These findings suggest that China, which has experienced diverse family forms, should strive to implement policies that facilitate family communication. For instance, public services, such as education and medical care, that are consistent with the local registered population should be provided in the permanent residence of migrant workers, which would allow left-behind children to live in cities with their parents. Moreover, the government introduced policies to prevent family breakdowns, such as a cooling-off period for divorce; however, this policy lacks sufficient flexibility and influence, and the effect is not satisfactory. The development of modern communication technology has brought people more convenient communication conditions and facilitated communication between family members. Social policy should be family oriented and systematically integrated.

The results of the present study have certain limitations. First, although the survey data used in this study are nationally representative, the sample age was mainly over 40 years old, and cannot reflect the situation of ACEs and depression among adults under 40 years old. Moreover, people born in the 1970s or 1960s experienced historical events, such as the Cultural Revolution and the transition from a planned economy to a market economy in their early childhood, which are unique and lack universal significance. Second, this study used a self-rated depression scale, which was not confirmed by a clinical evaluation by a licensed mental health provider and may have certain errors. Third, regarding the mechanism of ACEs and depression in adulthood, this study only examined the role of contact with children and did not include social support variables, such as contact with friends. Future studies can investigate additional moderator variables to identify more variables that can mitigate the effects of ACEs on depression in adults.

## 5. Conclusions

The present study explored the role of contact with children as a moderator in the effect of ACEs on depression in adulthood and found that enhancing contact with children could reduce the effects of at least two ACEs (parental drug abuse and starvation) on depression in adulthood.

These findings highlight the need to focus on the role of the family in individual mental health. First, we should develop a good family culture. As an important element of a social organism, the family carries out basic functions, such as providing guidance on social norms, moral education, cultural inheritance, and emotional comfort. Family culture has a direct, lasting, and subtle influence on the healthy growth of individual family members. Second, we must consolidate and cultivate family education in the youth. It is necessary to provide guidance around the implementation of the fundamental task of moral education and guide parents to use correct actions, thoughts, and methods to cultivate good thoughts, conduct, and habits in children. Family education is the foundation of all education, and it shoulders the important responsibility of cultivating values in young people. Finally, we should improve family services and promote social harmony. Harmony should be created between the society at large and small families, thus facilitating happiness and a sense of well-being in every family. Social harmony is ultimately reflected in the happiness of thousands of families and in the continuous improvement of the lives of hundreds of millions of people.

## Figures and Tables

**Figure 1 ijerph-19-08901-f001:**
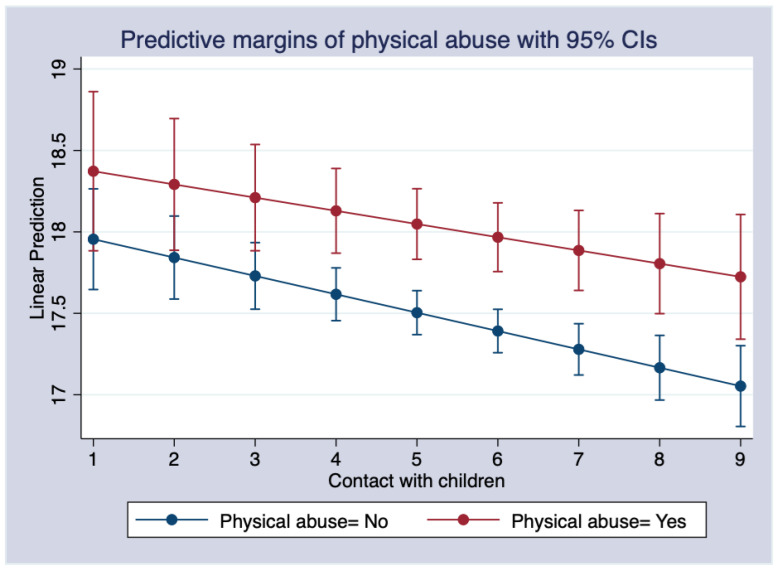
The marginal effects of contact with children on depression and physical abuse.

**Figure 2 ijerph-19-08901-f002:**
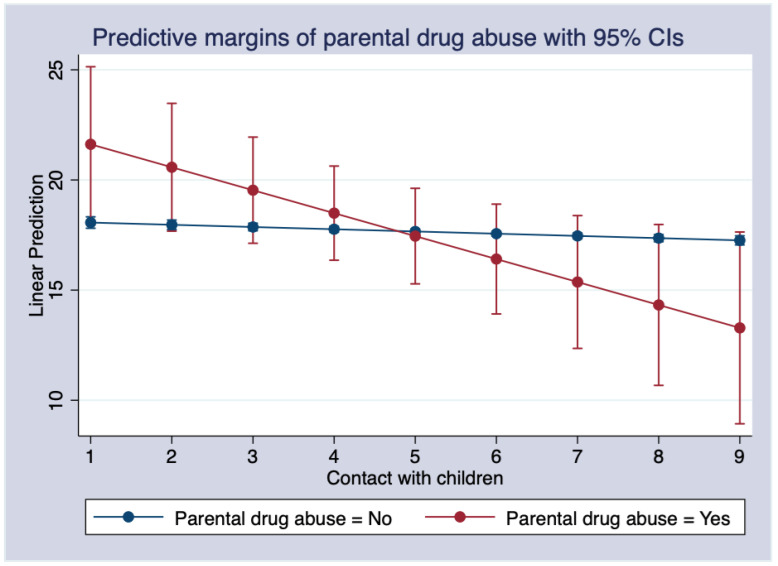
The marginal effects of contact with children on depression and parental drug abuse.

**Figure 3 ijerph-19-08901-f003:**
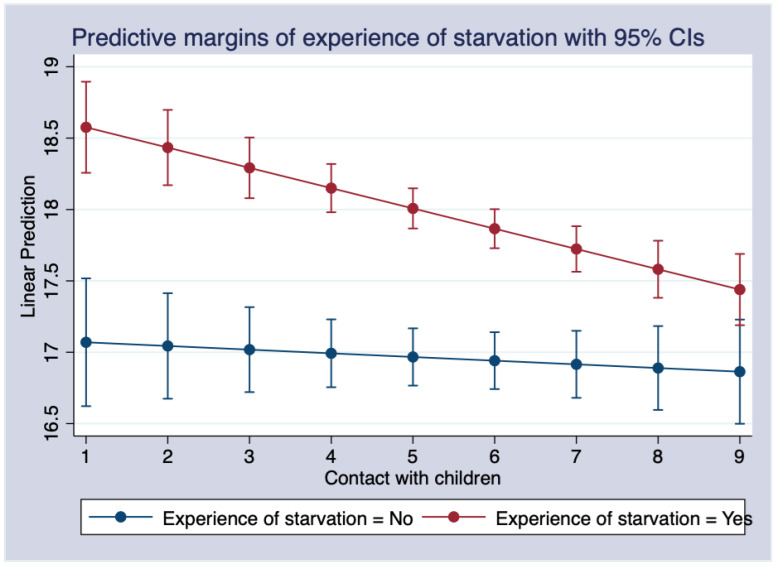
The marginal effects of contact with children on depression and experience of starvation.

**Table 1 ijerph-19-08901-t001:** The sample distribution of this study (N = 11,564).

Variables		Frequency	Percent (%)
Sex	Male	5501	47.57
	Female	6063	52.43
Ethnic	Minority ethnic groups	913	7.90
	Han ethnic groups	10,651	92.10
Hukou type	Rural	10,709	92.61
	Urban	855	7.39
Education level	Elementary and below	6672	57.70
	Middle school	3257	28.16
	High school	1312	11.35
	College or above	323	2.79
Religious belief	No religion	10,453	90.39
	Religious affiliation	1111	9.61
Political status	Non-party member	10,402	89.95
	Party member	1162	10.05

**Table 2 ijerph-19-08901-t002:** The depression scores by demographic characteristics (N = 11,564).

Categories		Depression Scores	
		(x ± s)	Test
Sex #	Male	16.604 ± 5.823	−16.504 ***
	Female	18.521 ± 6.593	
Hukou type #	Rural	17.750 ± 6.360	8.520 ***
	Urban	15.844 ± 5.368	
Education level +	Elementary and below	18.593 ± 6.632	148.77 ***
	Middle school	16.667 ± 5.733	
	High school	15.675 ± 5.259	
	College or above	14.628 ± 4.583	
Ethnic #	Minority ethnic groups	17.717 ± 6.150	0.542 †
	Han ethnic groups	17.599 ± 6.325	
Religious belief #	No religion	17.644 ± 6.313	−1.832
	Religious affiliation	17.279 ± 6.286	
Political status #	Non-party member	17.633 ± 6.324	−1.222
	Party member	17.394 ± 6.197	

Note: # Independent samples two-tailed *t*-test was used; + F-test was used; *** *p* < 0.001, † *p* < 0.1.

**Table 3 ijerph-19-08901-t003:** The depression scores with different ACEs by sex.

		Male		Female		All	
		(x ± s)	*t*-Test	(x ± s)	*t*-Test	(x ± s)	*t*-Test
Physical abuse	Yes	17.230 ± 6.122	−5.883 ***	19.656 ± 6.763	−7.516 ***	18.266 ± 6.514	−7.188 ***
	No	16.265 ± 5.626		18.168 ± 6.500		17.339 ± 6.206	
Emotional neglect	Yes	17.088 ± 6.057	−3.171 **	19.044 ± 6.793	−3.393 ***	18.164 ± 6.543	−5.032 ***
	No	16.476 ± 5.753		18.363 ± 6.524		17.452 ± 6.236	
Parental divorce	Yes	16.130 ± 5.268	0.390	20.531 ± 7.255	−1.729 †	18.691 ± 6.806	−1.275
	No	16.606 ± 5.825		18.510 ± 6.588		17.604 ± 6.309	
Parental death	Yes	17.581 ± 6.449	−4.373 ***	19.415 ± 7.036	−3.542 ***	18.506 ± 6.810	−5.239 ***
	No	16.483 ± 5.730		18.420 ± 6.534		17.504 ± 6.242	
Sibling death	Yes	17.338 ± 6.162	−5.025 ***	19.279 ± 6.930	−4.835 ***	18.360 ± 6.647	−6.892 ***
	No	16.393 ± 5.705		18.301 ± 6.476		17.392 ± 6.194	
Parental drug abuse	Yes	17.714 ± 7.376	−0.876	21.700 ± 9.129	−1.526	19.000 ± 8.053	−1.229
	No	16.599 ± 5.816		18.516 ± 6.588		17.605 ± 6.306	
Parental alcoholism	Yes	17.373 ± 6.348	−3.051 **	18.734 ± 6.495	−0.683	18.002 ± 6.449	−1.950 †
	No	16.529 ± 5.765		18.505 ± 6.600		17.576 ± 6.299	
Parental crime	Yes	20.304 ± 6.574	−3.057 **	19.000 ± 6.656	−0.272	19.811 ± 6.544	−2.126 *
	No	16.588 ± 5.815		18.520 ± 6.593		17.602 ± 6.310	
Parental illness/disability	Yes	18.233 ± 6.693	−10.413 ***	20.356 ± 6.987	−11.271 ***	19.376 ± 6.933	−15.397 ***
	No	16.201 ± 5.514		18.035 ± 6.398		17.156 ± 6.060	
Parental depression	Yes	18.808 ± 6.647	−14.486 ***	21.238 ± 5.910	−17.427 ***	20.125 ± 7.074	−22.737 ***
	No	16.038 ± 5.451		17.757 ± 6.191		16.931 ± 5.910	
Experience of starvation	Yes	17.061 ± 5.999	−9.069 ***	19.329 ± 6.818	−12.778 ***	18.192 ± 6.520	−14.235 ***
	No	15.512 ± 5.222		17.110 ± 5.923		16.433 ± 5.691	
Experience of violence	Yes	18.401 ± 6.316	−6.661 ***	20.568 ± 7.158	−7.842 ***	19.643 ± 6.891	−10.705 ***
	No	16.453 ± 5.755		18.308 ± 6.495		17.417 ± 6.220	
Being bullied by a schoolmate	Yes	18.497 ± 6.721	−9.906 ***	20.067 ± 7.014	−6.626 ***	19.239 ± 6.903	−10.708 ***
	No	16.290 ± 5.600		18.319 ± 6.509		17.369 ± 6.184	

Note: *** *p* < 0.001, ** *p* < 0.01, * *p* < 0.05, † *p* < 0.1.

**Table 4 ijerph-19-08901-t004:** The depression scores with different ACEs by Hukou type.

		Urban Hukou		Rural Hukou	
		(x ± s)	*t*-Test	(x ± s)	*t*-Test
Physical abuse	Yes	16.618 ± 5.533	−2.851 **	18.408 ± 6.573	−6.849 ***
	No	15.493 ± 5.259		17.482 ± 6.251	
Emotional neglect	Yes	16.521 ± 5.917	−2.127 *	18.313 ± 6.578	−4.850 ***
	No	15.620 ± 5.159		17.592 ± 6.289	
Parental divorce	Yes	17.750 ± 9.287	−0.711	18.765 ± 6.692	−1.143
	No	15.835 ± 5.351		17.745 ± 6.358	
Parental death	Yes	15.625 ± 5.127	0.362	18.687 ± 6.864	−5.278 ***
	No	15.864 ± 5.393		17.638 ± 6.288	
Sibling death	Yes	16.339 ± 5.691	−1.362	18.506 ± 6.688	−6.654 ***
	No	15.718 ± 5.280		17.529 ± 6.244	
Parental drug abuse	Yes	18.250 ± 5.560	−0.898	19.111 ± 8.436	−1.114
	No	15.833 ± 5.368		17.632 ± 6.276	
Parental alcoholism	Yes	17.000 ± 5.952	−1.747 †	18.075 ± 6.481	−1.545
	No	15.756 ± 5.315		17.722 ± 6.349	
Parental crime	Yes	18.000 ± 5.627	−1.067	20.233 ± 6.755	−2.142 *
	No	15.827 ± 5.366		17.743 ± 6.357	
Parental illness/disability	Yes	16.612 ± 6.233	−1.698 †	19.525 ± 6.938	−15.006 ***
	No	15.718 ± 5.206		17.280 ± 6.113	
Parental depression	Yes	17.469 ± 6.410	−3.475 ***	20.253 ± 7.080	−22.023 ***
	No	15.597 ± 5.152		17.050 ± 5.958	
Experience of starvation	Yes	16.442 ± 5.870	−3.087 **	18.289 ± 6.540	−13.004 ***
	No	15.312 ± 4.822		16.582 ± 5.781	
Experience of violence	Yes	18.446 ± 5.707	−3.781 ***	19.714 ± 6.952	−9.974 ***
	No	15.662 ± 5.300		17.560 ± 6.268	
Be bullied by schoolmate	Yes	15.630 ± 5.206	−3.118 **	19.384 ± 6.934	−10.266 ***
	No	17.358 ± 6.221		17.508 ± 6.235	

Note: *** *p* < 0.001, ** *p* < 0.01, * *p* < 0.05, † *p* < 0.1.

**Table 5 ijerph-19-08901-t005:** Results of OLS regression analysis (N  =  11564).

Variables	Model 1	Model 2	Model 3
Sex (Ref. = Male)	1.674 ***	1.841 ***	1.827 ***
	(0.123)	(0.120)	(0.120)
Ethnic (Ref. = Minority groups)	−0.215	−0.578	−0.592
	(0.215)	(0.222)	(0.222)
Hukou type (Ref. = Rural Hukou)	−0.898 ***	−0.255 *	−0.273 **
	(0.228)	(0.208)	(0.208)
Age	0.038 ***	0.033 ***	0.029 ***
	(0.006)	(0.006)	(0.006)
Religious belief (Ref. = No religion)	−0.391 *	−0.452 *	−0.460 *
	(0.197)	(0.190)	(0.190)
Political status (Ref. = Non-party member)	−0.766 ***	−0.757 ***	−0.726 ***
	(0.198)	(0.91)	(0.191)
Education level (Ref. = Elementary and below)			
Middle school	−1.317 ***	−0.965 ***	−0.944 ***
	(0.140)	(0.137)	(0.137)
High school	−2.133 ***	−1.701 ***	−1.664 ***
	(0.196)	(0.190)	(0.191)
College or above	−2.832 ***	−2.474 ***	−2.423 ***
	(0.366)	(0.356)	(0.356)
Physical abuse (Ref. = No)		0.565 ***	0.386
		(0.127)	(0.352)
Emotional neglect (Ref. = No)		0.430 ***	0.224
		(0.134)	(0.369)
Parental divorce (Ref. = No)		0.051	3.125 †
		(0.804)	(1.826)
Parental death (Ref. = No)		−0.023	0.847 †
		(0.184)	(0.495)
Sibling death (Ref. = No)		0.406 **	−0.091
		(0.134)	(0.371)
Parental drug abuse (Ref. = No)		0.505	4.493 *
		(1.077)	(2.153)
Parental alcoholism (Ref. = No)		0.128	0.607
		(0.207)	(0.573)
Parental crime (Ref. = No)		0.591	−0.682
		(0.985)	(2.845)
Parental illness/disability (Ref. = No)		1.237 ***	1.497 ***
		(0.141)	(0.394)
Parental depression (Ref. = No)		2.168 ***	2.615 ***
		(0.134)	(0.394)
Experience of starvation (Ref. = No)		0.958 ***	1.622 ***
		(0.123)	(0.333)
Experience of violence (Ref. = No)		0.991 ***	1.368 **
		(0.204)	(0.549)
Be bullied by schoolmate (Ref. = No)		1.230 ***	1.542 ***
		(0.170)	(0.477)
Physical abuse × Contact with children			−1.862 **
			(0.709)
Emotional neglect × Contact with children			−0.035
			(0.062)
Parental divorce × Contact with children			0.664 †
			(0.340)
Parental death × Contact with children			0.160 †
			(0.084)
Sibling death × Contact with children			−0.089
			(0.062)
Parental drug abuse × Contact with children			0.941 *
			(0.423)
Parental alcoholism × Contact with children			0.089
			(0.096)
Parental crime × Contact with children			−0.227
			(0.454)
Parental illness/disability × Contact with children			0.045
			(0.065)
Parental depression × Contact with children			0.081
			(0.065)
Experience of starvation × Contact with children			0.116 *
			(0.055)
Experience of violence × Contact with children			0.065
			(0.092)
Being bullied by schoolmate × Contact with children			0.057
			(0.079)
Constant	15.604 ***	13.746 ***	13.908 ***
	(0.443)	(0.439)	(0.537)
R^2^	0.055	0.116	0.120
F-value	74.45 ***	68.89 ***	43.45 ***
Df	9	22	36

Note: standard errors in parentheses; *** *p* < 0.001, ** *p* < 0.01, * *p* < 0.01, † *p* < 0.1.

## Data Availability

The datasets used and/or analyzed during the current study are available from the official website of CHARLS, http://charls.pku.edu.cn/index/zh-cn.html (accessed on 3 May 2022).

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
