# Peer review of "The Moderating Effect of Contact with Children on the Relationship between Adverse Childhood Experiences and Depression in Adulthood among a Chinese Adult Population"

_ijerph, 2022, doi:10.3390/ijerph19158901_

Round 1

Reviewer 1 Report

Comments to the authors

-        In the abstract, the manuscript mentioned that people with ACEs had significantly lower depression scores than those without ACEs. Is this right?

-        I think that the variable, contact with their children might be used as a moderating variable. If it is right, it should be mentioned that contact with their children used as a moderating variable in method part.

-        I am wondering why the study used depression only. In literature review, the impact of ACEs on anxiety is large.

-        Also, I am wondering how often contact with children means intergenerational communication. I think the title should be changed into the contact with children rather than intergenerational communication.

-        Discussion section needs to contain lots of rich arguments, which is very worthwhile. The points would read better with a clear that what are the summaries of the main findings, what research implications and suggestions for future research are discussed, and what kinds of program/policy implications can be made.

Author Response

Response to Reviewers

Dear  reviewer,

Thank you for your suggestions on this manuscript, and I have revised it according your comments. Express thankfulness for all.

 Reviewer 1:

  1. In the abstract, the manuscript mentioned that people with ACEs had significantly lower depression scores than those without ACEs. Is this right?

Response:Thank you for your comment, and I have revised as you suggested. We found that in categories such as physical abuse, emotional neglect, sibling death, parental illness/disability, parental depression, hunger, violence, and bullying, people with ACEs had significantly lower depression scores than those without ACEs. (See page 1, lines 15-18).

  1. I think that the variable, contact with their children might be used as a moderating variable. If it is right, it should be mentioned that contact with their children used as a moderating variable in method part.

Response: Thanks for your comment, and I agree with your suggestion. I added a new part in the methods section to point out contact with children used as a moderating variable, that is, “2.2.3. Moderating variable”. (See page 4, lines 147-154).

  1. I am wondering why the study used depression only. In literature review, the impact of ACEs on anxiety is large.

Response:Thank you for your comment. Indeed, in the existing literature, many studies have analyzed the effects of ACEs on anxiety, and some scholars have also studied the effects of ACEs both on anxiety and depression. In addition, many studies have also analyzed the effect of ACEs on depression. For example, Von Cheong, E., Sinnott, C., Dahly, D., & Kearney, P. M. (2017). Adverse childhood experiences (ACEs) and later-life depression: perceived social support as a potential protective factor. BMJ open, 7(9), e013228; Chapman, D. P., Whitfield, C. L., Felitti, V. J., Dube, S. R., Edwards, V. J., & Anda, R. F. (2004). Adverse childhood experiences and the risk of depressive disorders in adulthood. Journal of affective disorders, 82(2), 217-225.

The reason why we chose to study the effect of ACEs on depression is that the well-established depression questionnaire in the CHARLS questionnaire was used to investigate the depression of respondents, while the measurement of anxiety only used a few questions and did not use existing well-established scales to investigate. This may be the limits of this study.

  1. Also, I am wondering how often contact with children means intergenerational communication. I think the title should be changed into the contact with children rather than intergenerational communication.

Response:Thanks for your comment, and I agree with your suggestion. I changed intergenerational communication to contact with children. (see page 2, line 55;page 2, line 96;page 12, line 335,339,340,344).

  1. Discussion section needs to contain lots of rich arguments, which is very worthwhile. The points would read better with a clear that what are the summaries of the main findings, what research implications and suggestions for future research are discussed, and what kinds of program/policy implications can be made.

Response:Thank you for your comment, and I have revised as you suggested.

At the beginning of the discussion section, we summarize our findings from three aspects. That is, “This study examined the effect of ACEs on depression and the moderating role of contacting with children based on CHARLS data. First, this study further confirmed the fact that most ACEs contribute to depression in adulthood. The present study examined 13 types of ACEs. Among the 13 ACEs, eight were significant, accounting for 62%. More specifically, adults who experienced physical abuse, emotional neglect, sibling death, parental illness/disability, parental depression, the experience of starvation, the experience of violence, and being bullied by schoolmates in childhood were more likely to experience depression in adulthood than those who did not. This finding was consistent with extant studies [40–42].

Second, the current study demonstrated that not all ACEs cause depression in adulthood and further illustrated that different ACEs have different outcomes. Although the ever-increasing literature indicates that ACEs may trigger mental health problems, such as depression [2,4], not all individuals who experienced ACEs develop depression. Five ACEs (parental divorce, parental death, parental drug abuse, parental alcoholism, and parental crime) did not cause depression in adulthood. However, this does not mean that these ACEs did not cause other harm. For instance, parental divorce has been shown to lead to lower self-rated health [43]. The reasons and mechanisms behind only certain ACEs leading to depression and not others require further research.

Third, this study determined the role of contacting with children in mitigating the effects of ACEs on depression risks in adulthood. The results revealed that an increase in contact with their children resulted in lower depression scores in adulthood among individuals who experienced parental drug abuse and starvation in childhood, indicating that contact with children mitigated these adverse effects. Childhood adversity should be healed in adulthood, and the family is an important arena for such healing. Contact with children helps make ACE victims experience the family warmth that they did not experience as children, thereby reducing the risk of depression. Starvation is a physical experience, and it develops into a psychological experience as an individual grows, evolving into a fear of starvation and a sense of insecurity. Contact with children enables individuals suffering from hunger to gain a sense of security through interaction with family members.” (See pages 11-12, lines 485-519).

Furthermore, we further elaborate on the implications for family policy of research findings. That is, “this sheds light on the importance of family policy in reducing the negative impact of childhood adversity [44–46]. These findings suggest that China, which has experienced diverse family forms, should strive to implement policies that facilitate family communication. For instance, public services, such as education and medical care, that are consistent with the local registered population should be provided in the permanent residence of migrant workers, which would allow left-behind children to live in cities with their parents. Moreover, the government introduced policies to prevent family breakdowns, such as a cooling-off period for divorce; however, this policy lacks sufficient flexibility and influence, and the effect is not satisfactory. The development of modern communication technology has brought people more convenient communication conditions and facilitated communication between family members. Social policy should be family-oriented and systematically integrated.” (See page 12, lines 533-544).

Finally, in the part of limitations, we further discuss suggestion for future research. That is, “regarding the mechanism of ACEs and depression in adulthood, this study only examined the role of contacting with children and did not include social support variables, such as contact with friends. Future studies can investigate additional moderator variables to identify more variables that can mitigate the effects of ACEs on depression in adults.” (See page 12, lines 552-556).

Reviewer 2 Report

Dear authors, thank you for giving me the opportunity to review your manuscript: “The moderating effect of intergenerational communication on the relationship between adverse childhood experiences and depression in adulthood”. 

I think that this manuscript can contribute to the literature. 

I send below some comments to improve the manuscript.

Title:

-        I think the title should have reference to the Chinese population.

Introduction:

-        Authors should add more recent references to the introduction.

-        The authors start by referring to the impact of ACEs in the first paragraph; in the second paragraph, they refer to the Chinese reality concerning the family reality, and then they refer to the impact of ACEs again. This happens without a thread. I think the authors should invest in showing a thread in the introduction that meets the study's objectives. For example, in the first paragraph, they write: "In addition, some meta-analyses confirm that a history of physical or sexual abuse in childhood significantly increases the risk of depression and anxiety in adulthood" and in the third paragraph, they write, "Extensive empirical evidence indicates that ACEs cause mental health problems across the lifespan of an individual.". They repeat some information, and I think the introduction should be more structured.

-        Authors refer: “However, the mechanisms and mitigating factors of ACEs on depression in adults remain unclear. In particular, the bridging effect of communication on mental health disadvantages caused by ACEs has not yet been established. Identifying this effect could facilitate a better understanding of the role of ACEs on adult mental health outcomes”. 

o   They should indicate if there are any studies on mechanisms and mitigating factors of ACEs and, if so, what they refer to?

o   Several studies indicate, for example, the mitigating effect of benevolent childhood experiences. Others indicate other factors. The authors should elaborate on this question, specifying the existence or not of studies that explore intergenerational communication.

Materials and Methods: 

-        Please provide details about the inclusion criterium. 

-        Please provide more information about participants (e.g., age, gender, ...)

Discussion

-         Authors should explain the results “Furthermore, this study determined the role of intergenerational communication in mitigating the effects of ACEs on depression risks in adulthood.” with the literature review. That is one of the main goals of the article, and the authors do not refer (in the introduction or discussion) to how the literature explains this.

o   Do these results match the literature? 

o   According to the literature, why do you have these results?

Author Response

Response to Reviewers

Dear reviewer,

Thank you for your suggestions on this manuscript, and I have revised it according your comments. Express thankfulness for all.

Reviewer 2:

  1. Title: I think the title should have reference to the Chinese population.

Response:Thank you for your comment, and I have revised as you suggested. I mentioned the Chinese population in the title. That is, I change the title to “The moderating effect of contacting with children on the relationship between adverse childhood experiences and depression in adulthood among Chinese adult population”.

  1. Introduction:

(1) Authors should add more recent references to the introduction.

Response: Thank you for your comment, and I have added recent publications. For example,

Abbott, M.; Slack, K. S. Exploring the relationship between childhood adversity and adult depression: A risk versus strengths-oriented approach. Child Abuse & Neglect 2021, 120, 105207.

Masiano, S. P.; Yu, X.; Tembo, T.; Wetzel, E.; Mphande, M.; Khama, I.; et al. The relationship between adverse childhood experiences and common mental disorders among pregnant women living with HIV in Malawi. Journal of Affective Disorders 2022, 312, 159–168.

Chen, H.; Fan, Q.; Nicholas, S.; Maitland, E. The long arm of childhood: The prolonged influence of adverse childhood experiences on depression during middle and old age in China. Journal of Health Psychology, 2021. https://doi.org/10.1177/13591053211037727

Jiang, W.; Ji, M.; Chi, X.; Sun, X. Relationship between adverse childhood experiences and mental health in Chinese adolescents: Differences among girls and boys. Children 2022, 9(5), 689.

Cavanaugh, C.; Nelson, T. A national study of the influence of adverse childhood experiences on de-pression among Black adults in the United States. Journal of Affective Disorders 2022, 311, 523–529.

(2) The authors start by referring to the impact of ACEs in the first paragraph; in the second paragraph, they refer to the Chinese reality concerning the family reality, and then they refer to the impact of ACEs again. This happens without a thread. I think the authors should invest in showing a thread in the introduction that meets the study's objectives. For example, in the first paragraph, they write: "In addition, some meta-analyses confirm that a history of physical or sexual abuse in childhood significantly increases the risk of depression and anxiety in adulthood" and in the third paragraph, they write, "Extensive empirical evidence indicates that ACEs cause mental health problems across the lifespan of an individual.". They repeat some information, and I think the introduction should be more structured.

Response:Thank you for your comment, and I have revised as you suggested. I have restructured the "Introduction" section and put the part of Chinese context at the end the section. Meanwhile, I have removed redundant information. That is, “In China, children’s experiences are gradually being diversified owing to social transformation. In 2021, the number of rural migrant workers in cities reached 290 million [29]. Left-behind children in rural areas are often looked after by their grandparents, whereas some children move to cities with their parents and face a life of displacement [30–31]. Furthermore, the family structure in China is changing, and the divorce rate is increasing annually. The crude divorce rate rose from 1.1% in 1987 to 3.1% in 2020 [32–33]. Moreover, as the divorce rate rises, the phenomenon of remarriage also increases. Therefore, data show that the remarriage rate after divorce reached 60%, and the divorce rate after remarriage was as high as 80% in 2020 [33]. Evidently, the proportion of children who suffered from family changes increased, and it was found that children in single-parent households or restructured families were more likely to suffer negative family experiences [34]. Thus, as Chinese families diversify, the effect of ACEs on adulthood becomes particularly important.” (See page 1-2, lines 25-199).

(3) Authors refer: “However, the mechanisms and mitigating factors of ACEs on depression in adults remain unclear. In particular, the bridging effect of communication on mental health disadvantages caused by ACEs has not yet been established. Identifying this effect could facilitate a better understanding of the role of ACEs on adult mental health outcomes”.

  1. They should indicate if there are any studies on mechanisms and mitigating factors of ACEs and, if so, what they refer to?
  2. Several studies indicate, for example, the mitigating effect of benevolent childhood experiences. Others indicate other factors. The authors should elaborate on this question, specifying the existence or not of studies that explore intergenerational communication.

Response:Thank you for your comment, and I have revised as you suggested. First, I have added the relevant literature on protective factors and mentioned relevant studies on the mechanisms and mitigating factors of ACEs. That is, “Some scholars have discussed the influencing mechanisms and factors alleviating the effect of ACEs as many individuals experience ACEs without developing depression and other psychological problems. Many studies have identified multiple factors that reduce the effect of ACEs on depression. For example, some scholars have found that positive childhood experiences at home and school protect at-risk adolescents from mental health problems [22]. Mindfulness is a cognitive resource that protects people from symptoms of psychological distress [23]. Self-esteem mediates the negative correlation between child maltreatment and depressive symptoms, and protects against the adverse effects of early adversity exposure on mental health [24]. Researchers have also found that resilience moderated the association between ACEs and depression and that the association between ACEs and depression was stronger in low-resilient individuals than in high-resilience individuals [25]. Some scholars have found that religious service attendance and ethnic identity are protective factors for the effects of ACEs on depression [21]. Moreover, some studies revealed that the effect of ACEs on depression may be reduced by managing current stressors and building psychological resilience [26]. Interventions that focus on improving emotion-regulation skills might provide an efficient “transdiagnostic” treatment strategy for psychological and physical health problems [27]. A longitudinal study on acculturation stress and the relationship between ACEs and depression in a sample of young adult Hispanic immigrants indicated that ACEs predicted depressive symptoms and revealed significant mediating and moderating effects of cumulative and distinct facets of acculturation stress [28].” (See page 2, lines 150-170).

Second, we discussed studies concerning the mitigating effects of positive childhood experiences of parents. That is, “Some studies have examined the effects of intergenerational transmission of ACEs and found that higher parental childhood adversity scores were associated with higher children's depression, [35] while parents' positive childhood experiences counteract intergenerational ACEs [36]. However, the bridging effect of intergenerational communication on mental health disadvantages caused by ACEs has not yet been established.” (See page 2, lines 187-192).

  1. Materials and Methods:

Please provide details about the inclusion criterium. Please provide more information about participants (e.g., age, gender, ...).

Response: Thank you for your comment, and I have revised as you suggested. First, in this revision, I specified the age range of the respondents in the sample. That is, “The average age of respondents was 57.49, with the youngest being 30 and the oldest being 84.” (See page 13, line 115-116). Second, I added a table (Table 1) that describes the demographic and socioeconomic characteristics of the participants. (See page 3, table 1).

  1. Discussion

Authors should explain the results “Furthermore, this study determined the role of intergenerational communication in mitigating the effects of ACEs on depression risks in adulthood.” with the literature review. That is one of the main goals of the article, and the authors do not refer (in the introduction or discussion) to how the literature explains this. Do these results match the literature? According to the literature, why do you have these results?

Response: Thank you for your comment.

First, I included relevant literature on contacting with children in the “Introduction” section. That is, “Some studies have examined the effects of intergenerational transmission of ACEs and found that higher parental childhood adversity scores were associated with higher children's depression, [35] while parents' positive childhood experiences counteract intergenerational ACEs [36].” (See page 2, lines 187-190).

Second, in the “Discussion” section, we discuss the findings of this study with the conclusions of the existing literature. That is, “First, this study further confirmed the fact that most ACEs contribute to depression in adulthood. The present study examined 13 types of ACEs. Among the 13 ACEs, eight were significant, accounting for 62%. More specifically, adults who experienced physical abuse, emotional neglect, sibling death, parental illness/disability, parental depression, the experience of starvation, the experience of violence, and being bullied by schoolmates in childhood were more likely to experience depression in adulthood than those who did not. This finding was consistent with extant studies [40–42].” (See page 11, lines 485-493).

Also, “Some scholars have noted that resilience [25] and stress management [26] as moderating variables are particularly important in reducing the effect of ACEs on depression. However, resilience and stress management are both intrinsic qualities of the individual, and do not reveal how family intergenerational relationships affect people ‘s mental health. This study found that contact with children moderates the relationship between ACEs and depression, further underscoring the role of family intergenerational relationships in reducing the vulnerable effect of ACEs.” (See page 12, lines 519-525).

Finally, in the “Discussion” section, we explain the findings of this study. That is, “The effects of ACEs on adult depression further indicated the importance of family support. With the moderation of contact with children, parental divorce and death had significant marginal effects. Parental divorce and death are family issues. As mentioned above, Chinese families are becoming increasingly diverse with rising divorce rates and an increase in single-parent families. The results of this study demonstrated that better communication with their children could reduce the impact of parental divorce and death on depression in an individual’s adult life. In other words, pain caused by the family should be treated by the family. Furthermore, this sheds light on the importance of family policy in reducing the negative impact of childhood adversity [44–46]. These findings suggest that China, which has experienced diverse family forms, should strive to implement policies that facilitate family communication. For instance, public services, such as education and medical care, that are consistent with the local registered population should be provided in the permanent residence of migrant workers, which would allow left-behind children to live in cities with their parents. Moreover, the government introduced policies to prevent family breakdowns, such as a cooling-off period for divorce; however, this policy lacks sufficient flexibility and influence, and the effect is not satisfactory. The development of modern communication technology has brought people more convenient communication conditions and facilitated communication between family members. Social policy should be family-oriented and systematically integrated.” (See page 12, lines 526-544).